# Do Emotion Regulation Strategies Mediate the Relationship of Parental Emotion Socialization with Adolescent and Emerging Adult Psychological Distress?

**DOI:** 10.3390/healthcare11192620

**Published:** 2023-09-25

**Authors:** Liliana Bujor, Maria Nicoleta Turliuc

**Affiliations:** 1Faculty of Sciences of Education, Ştefan cel Mare University of Suceava, 720229 Suceava, Romania; 2Department of Psychology, Alexandru Ioan Cuza University, 700554 Iasi, Romania; turliuc@uaic.ro

**Keywords:** parental style of emotion socializing, cognitive reappraisal, expressive suppression, psychological distress, anger, happiness

## Abstract

A child’s ability to cope with stress is shaped by experiences in a parent–child relationship. In this study, the direct effect of a parent’s response to anger and happiness in childhood on adolescents’ and emerging adults’ psychological distress and the indirect effect through the mediating role of emotion regulation strategies—specifically, cognitive reappraisal and emotional suppression—were measured. To achieve our research aim, we tested four parallel mediation models using the bootstrapping method. A group of 497 participants aged between 14 and 35 years (*M* = 18.62; *SD* = 3.32), 66% female (*n* = 332) and 34% male (*n* = 165), completed a questionnaire comprising self-reporting measures. The results indicate direct effects between emotion socialization and distress for seven independent variables. The mother’s and father’s positive responses to anger and happiness are significant negative predictors of distress; the negative responses of both parents to happiness, and the mother’s negative response to anger—but not the father’s—are significant positive predictors of distress. The findings also provide support for the mediating role of expressive suppression and cognitive reappraisal for the mother’s positive response to both anger and happiness, as well as for the mother’s negative response to the child’s expression of happiness. None of the father’s responses—positive or negative, in relation to anger or happiness—are mediated by emotion regulation strategies in relation to distress. Our findings have practical implication for a preventative intervention program focused on the psychological growth of adolescents by adaptative emotional responses.

## 1. Theoretical and Conceptual Framing

Developmental studies consistently indicate that parents have a primary role in shaping a child’s emotional development through their direct and indirect, verbal and nonverbal messages addressed to their children. Dealing with anger, happiness, fear or sadness are emotional and social daily lessons that put parent and child together in a positive or negative interaction with implications for their development and wellbeing. 

Our conceptual framework is the Tripartite Model of the Impact of the Family on Children’s Emotion Regulation and Adjustment [1]. Family context affects the development of emotion regulation in three ways: (1) observation via social modeling, (2) specific parenting practices in relation to emotions and (3) the emotional climate of the family, reflected in attachment, parenting style and marital relations. The family context has direct effects on children’s adjustment (e.g., internalizing, externalizing), but much of the effects of the family context are mediated through children’s emotion regulation [2].

According to the tripartite model of parental influences [1], the family context impacts emotional development through three pathways: the emotional climate, the parenting style and the emotional quality of marital relationships. Parent–child interactions with all its components (parents’ reactions to the child’s emotions), whether supportive (e.g., *reward*) or unsupportive (e.g., *punishment*, *neglect*), are reflected in their emotional life and represent an important predictor for the development of emotion regulation [3] and wellbeing [4]. Thus, parental emotion socialization is a process that helps a child to identify and appropriately express and manage their emotions, due to parental reactions to a child’s emotions. Certain retrospective reports of adolescents have shown that parental socialization emotional strategies project emotional effects into adulthood. The Malatesta-Magai model of the parental style of emotion socialization [5], which defined the concepts and variables of this study, delineates five distinct strategies used by parents when it comes to emotion socialization: *rewarding (i.e.*, *provision of comfort and empathy following children’s emotion expression)*, *punishing (i.e.*, *discouraging or punishing expression)*, *overriding (i.e.*, *suggesting that others have it worse or distracting children from emotion)*, *neglecting (i.e.*, *ignoring emotional expression) and magnifying (i.e.*, *parent matches the child’s expression of emotion equally or with more intensity)*. 

From a functionalist approach, emotion socialization implies responses to concrete emotions. In this research, we will analyze anger and happiness because a number of studies have identified the existence of core emotions relevant for emotional development, frequently implied in internalization or externalization problems [6,7]. Anger is an emotion that communicates a need for limits and rules and activates a defense system. Happiness functions as a signal to engage in activities that bring personal satisfaction, promote positive relationships through emotional contagion and ensure wellbeing [7]. 

According to a processual model of emotion regulation [8], there are many strategies that can intervene in different moments of emotional experience: *anterior-focused*, like situation selection, situation modification, attentional deployment and cognitive change; or *response-focused*, which can be response modulation. A specific type of cognitive change is *cognitive reappraisal* (CR), and for response modulation, there is *expressive suppression* (ES). CR and ES are two strategies with multiple implications for mental health and wellbeing.

A person who activates CR tends to negotiate stressful events by interpreting them in an optimistic manner [9,10] and has a high level of life satisfaction and self-esteem, as well as a lower level of anxiety, depression or posttraumatic stress disorder [3,11,12]. ES involves the inhibition of emotion expression and leads to a series of psychological consequences, like both externalizing and internalizing problems in early childhood, through adolescence and emerging adulthood [13]. 

Although together, all these theories and models explain the emotional impact of parents with regard to their child’s development; it is important to extend knowledge by examining the role of emotion regulation strategies, like protective factors, between parental influences and distress [14].

A principal limitation of specific studies on the emotion socialization process in relation to emotion regulation, stress and mental health is that only the mother is included in reports. The lack of inclusion of the father is a significant limitation to a deep understanding of parenting influences and effects [15], and efforts are currently proposed to evidence the implication of other caregivers, like fathers [16,17,18,19,20]. Another distinctive addition to the literature is our focus on discrete emotions, anger and happiness, in parents’ emotion socialization. Our examination of emotion socialization relied on two composite variables, both for negative and positive parenting behaviors in relation to one negative and one positive discrete emotion (anger and happiness). In the literature, there are few studies that analyze discrete emotions (sadness and anger) in relation to parental emotion socialization [6]. 

The relationship between our variables were studied more in Western, individualistic cultures and less in Eastern, collectivist ones. Because of the fact that, in the last twenty years, there has been a paradigm change in Eastern culture from an authoritarian parenting style to an authoritative parenting style, which emphasizes children’s wellbeing and less distress [21], it is important to understand the explanatory mechanisms between different parental styles as relates to emotion socialization and distress in an Eastern cultural context. 

In accordance with these limitations and recommendations, this study expands the existing literature in three important ways: first, we integrated fathers’ responses to children’s emotions, obtained different analyses for both parents and highlighted the contribution of fathers to the emotion socialization process; secondly, we explored two discrete emotions for a specific understanding of the emotion socialization process; thirdly, we proposed an explanatory mechanism in the relationship between parental strategies for emotion socialization during childhood and distress in adolescence and emerging adulthood. 

Because of the evidence from the literature about long-term parental influences [11,22], we decided to analyze longer periods after childhood, in particular adolescence and emerging adulthood (a period that is neither adolescence nor young adulthood, from 18 to 25 years and normative in adulthood but not having entered a phase concerned with enduring responsibilities) [23]. 

The purpose of this study is to have a general view of the relation between maternal and paternal emotion socialization of anger and happiness in the childhood period and distress in adolescence and emerging adulthood. 

## 2. Problem Statement

### 2.1. Parental Emotion Socialization and Its Emotional Consequences

Significant correlations between negative emotion socialization and internalization issues are a constant in several studies; thus, sadness or fear and punishment or neglect were associated with high levels of psychological distress in adulthood [7,24,25]. Punishment of positive emotions correlates with high levels of distress, while reward is associated with lower levels of distress [22].

Although the topic of emotion socialization is important and relevant in multiple areas of psychology, few studies have examined the socialization process of specific negative (e.g., fear, anger) or positive emotions (e.g., happiness) as separate influences stemming from both the mother and the father [4]. Furthermore, there is relatively extensive research on negative emotions and a lack of research on positive socialization emotions [22]. 

There are a few studies regarding emotion socialization as separate influences from the mother and the father [26]. Although there are concordant findings suggesting that a mother’s focus and implication is deeper in relation to their child’s emotions than the father’s [6,13,25,27], the data regarding the long-term implications of the mother and father’s emotional behavior produced mixed results. A meta-analysis highlighted that paternal psychopathology is more related to children’s emotional and behavioral problems than maternal psychopathology [28], and paternal emotion socialization is more consistently related to the psychopathology of daughters [22]. A father’s response to their child’s emotions has a powerful relation to the child’s emotional skills [27]. More precisely, a father’s acceptance attitude towards their child’s sadness and anger at five years old is associated with better social skills at eight years old [29]. Based on this finding, we formulated the first objective of this research: to analyze the relation between anger and happiness socialization in childhood and their effects on adolescents and emerging adults’ distress by conducting a separate analysis for both the mother and the father. 

### 2.2. Emotion Regulation Strategies as Mediators

Recent research [3,13,30,31] investigated the mediating role of emotion regulation strategies between parental responses to a child’s emotions and emotional consequences in adolescence and adulthood, but there is a lack of evidence concerning specific analyses, like separate investigations for mothers and fathers in the case of specific emotions (e.g., anger, fear, happiness, sadness). For example, in a family with different maternal and paternal nonconsensual parental responses to a specific emotion, it is difficult to build coherent emotional behavior and adaptative emotion regulation strategies [13]. The quality of parent–child interactions has profound implications for the expression and regulation of emotions that can become a mediating variable between parental influences and distress or wellbeing [13,32]. In this context, recent research has shown interest in studying responsible mechanisms when it comes to the relation between parental influences and emotional development [3] and suggests the role of emotion regulation as a mediating variable be studied. Summarizing these findings, we established the second objective: to analyze the relation between maternal and paternal emotion socialization in childhood and their effects on adolescent and emerging adult distress, taking into account the mediational role of both parents’ cognitive reappraisal and expressive suppression as emotion regulation strategies. 

## 3. This Study

Due to the impact of emotional experiences during childhood on mental health [3,13,33], it is useful to investigate the mediator’s role of emotion regulation between emotional experiences during childhood and emotional life during adolescence or early adulthood. Based on the tripartite model of family relationships [1], related with the Malatesta-Magai model of the parental strategies of emotion socialization [5], a functionalist approach of emotions and the processual model of emotion regulation [8], we tested four mediation models in order to answer the question of whether emotion regulation strategies can mediate positive and negative parental impacts in childhood—separate for the mother and father—regarding distress in adolescence and emerging adulthood in relation to two distinct emotions: anger and happiness. 

In our research, family context was represented by parental practices of emotion socialization in childhood that was then related to adolescent adjustment as measured by the level of distress, like internalizing difficulty, via emotion regulation strategies (cognitive reappraisal and emotion suppression). Each of these concepts was integrated in a distinct theoretical model; parental emotion socialization was related with The Malatesta-Magai Model of the Parental Style of Emotion Socialization [5], and emotion regulation was explained from a Processual Model of Emotion Regulation perspective [8]. This conceptual framework, globally and sequentially presented, was analyzed from a functionalist approach that implies responses to concrete emotions (e.g., happiness, anger). Together, this conceptual framework and this theoretical model became measures in our research investigation: The Emotion as a Child Scale—EAC, Version 2 [7] and The Emotion Regulation Questionnaire—ERQ [8]. 

Therefore, the following hypotheses were formulated: 

**Hypothesis 1.** 
*The relation between negative anger socialization (mother and father) and distress is mediated by emotion regulation (expressive suppression, cognitive reappraisal).*


**Hypothesis 2.** 
*The relation between positive anger socialization (mother and father) and distress is mediated by emotion regulation (expressive suppression, cognitive reappraisal).*


**Hypothesis 3.** 
*The relation between negative happiness socialization (mother and father) and distress is mediated by emotion regulation (expressive suppression, cognitive reappraisal).*


**Hypothesis 4.** 
*The relation between positive happiness socialization (mother and father) and distress is mediated by emotion regulation (expressive suppression, cognitive reappraisal).*


## 4. Materials and Methods

### 4.1. Participants 

Initially, 525 participants accepted our invitation to fill out the research questionnaire. By controlling the variable *type* of *family origin*, we only chose the questionnaires from respondents who come from intact families. We excluded 28 questionnaires from participants who reported having one biological parent and one step-parent or as belonging to a single-parent family. The final sample consisted of 497 participants, aged between 14 and 25 years (*M* = 18.62; *SD* = 3.32), of whom 53% were adolescents, 47%—emerging adults, 66%—female (*N* = 332) and 34%—male (*N* = 165). The sample is composed of students from Stefan cel Mare University of Suceava (55%) as well as high school students (45%) from several national state colleges (Table 1). 

### 4.2. Procedures

First, the study protocol was approved by the Ethics Committee of Alexandru Ioan Cuza University (1013/17.05.2022). Two high schools agreed to take part in the study and to inform the classes of students aged between 14 and 18 years and one university with students aged between 19 and 25 years. Second, all the participants received and signed their informed consent. Then, the participants filled out the questionnaire, which included three scales and several sociodemographic items (*gender*, *age*, *school level*, *family type*). The data were collected using self-reporting scales, applied in pencil-paper format, in different educational contexts: at courses, seminars and classes and in the presence of one of the research team members. Time for the administration of the questions was 25–30 min for all the questionnaires. To avoid social desirability, the questionnaires were anonymous (except for the participants who wanted personal results). 

### 4.3. Measures 

#### 4.3.1. Parental Emotion Socialization

Emotion as a Child Scale—EAC (Version 2) [7] was measured in the following way: For each scale, the participants recalled parental responses (separately for the mother and father) to their positive (e.g., happiness) or negative (e.g., anger) emotions during their childhood period. The instrument had 15 items, evaluated on a Likert Scale in five steps from 1 (*not at all*) to 5 (*very much*), for each analyzed emotion and the following five parental styles: Reward (item: *Helped me to deal with the issue*), Punish (item: *Gave me a disgusted look*), Override (item: *Bought me something I liked*), Neglect (item: *Ignored me*) and Magnify (item: *Got anxious herself/himself*). Due to data reduction strategies, we computed and used two summary indexes: positive parental emotion socialization (Reward scale) and negative parental emotion socialization (Neglect and Punishment scales). Factor analyses with the negative emotion scales in this study demonstrated that Punishment and Neglect are the most representative for negative parental emotion socialization styles. In this study, Cronbach’s alphas were between 0.70 and 0.89 for all dimensions.

#### 4.3.2. Emotion Regulation

The Emotion Regulation Questionnaire—ERQ, [8] had 10 items and measured cognitive reappraisal (item: *When I want to feel more positive emotions*, *I change the way I’m thinking about the situation*) and expressive suppression (item: *When I am feeling negative emotions*, *I make sure not to express them*). Answers were scored on a 7-point Likert Scale from 1 (*strongly disagree*) to 7 (*strongly agree*). The cognitive reappraisal subscale had an adequate internal consistency with a Cronbach’s alpha = 0.76, and for expressive suppression, we had a Cronbach’s alpha = 0.75. 

#### 4.3.3. Psychological Distress

The Profile of Emotional Distress—PED, [34] is a 26-item scale that measured dysfunctional negative emotions and functional negative emotions in the *fear (e.g.*, *Anxious*, *Panicked*) and *sadness/depression (e.g.*, *Sad*, *Hopeless*) categories. The scale was designed in 2005, starting from the Profile of Mood Disorders, a Short Version [35]. The scale allowed for the calculation of a general distress score by adding item scores evaluated by the Likert Scale from 1 (*not at all*) to 5 (*extremely*). The Psychological Distress Scale had an adequate internal consistency with a Cronbach’s alpha = 0.95.

### 4.4. Data Analyses

To conduct our analyses, we used a version 2.3.16 JOMOVI GLM Mediation Model. We calculated both the direct and indirect effects of positive and negative parental emotion socialization (mother/father; anger/happiness) on psychological distress through emotion regulation strategies (cognitive reappraisal and expressive suppression). This was achieved by testing the independent variables (anger positive socialization by the mother, anger positive socialization by the father, anger negative socialization by the mother, anger negative socialization by the mother, happiness positive socialization by the mother, happiness positive socialization by the father, happiness negative socialization by the mother and happiness negative socialization by the father), dependent variables (psychological distress) and mediators (cognitive reappraisal and expressive suppression) in four parallel mediation models, while checking for key background variables (sex, age and family structure). We used 5000 bootstrapped samples, and the biases were corrected at 95% confidence intervals (CI) for each indirect effect, where the significance of indirect effect path was indicated when the confidence interval did not contain zero (*p* < 0.05). Bootstrapping is a resampling method that constructs a confidence interval around the examined indirect effect and provides a more accurate estimate of indirect effects independently via sample distribution (normal or not) [36,37].

## 5. Results

Descriptive statistics (means, standard deviation, minimum, maximum, Skewness and Kurtosis) are presented in Table 2. 

Bivariate correlations among the main constructs are reported in Table 3. All positive answers by the mother and father for anger and happiness have significant and negative correlations with psychological distress. Negative emotion socialization has significant and positive correlations with psychological distress. Paternal reward of anger does not have a correlation with expressive suppression, and paternal negative socialization of anger does not have a significant correlation with cognitive reappraisal. 

According to our research in the analysis of the relation between parental emotion socialization in childhood and its effects in adolescence and emerging adulthood with regard to distress, and taking into account a separate analysis for the mother and father as well as specific emotions (anger and happiness), we have significant results (Table 4).

As we expected, the coefficients for the direct effect show that anger positive socialization (rewarding) is negatively associated with distress for the mother (b = −0.83, *p* < 0.05) and the father (b = −0.68, *p* < 0.05). A negative socialization of anger is a significant predictor for distress in relation to the mother’s response (b = 0.91, *p* < 0.001) but not the father’s.

In relation to happiness, the father’s positive (b = −0.85, *p* < 0.05) and negative (b = 0.65, *p* < 0.05) responses, and the mother’s negative (b = 0.54, *p* < 0.05) responses are significant predictors for distress. 

The parental response is significant, not for distress, but for emotion regulation strategies too. Both cognitive reappraisals and expressive suppression are predicted just from the mother’s response. Anger positive socialization (b = 0.56, *p* < 0.001) is a significant predictor for cognitive reappraisal, while happiness positive (b = 0.53, *p* < 0.001) and negative (b = −0.29, *p* < 0.001) responses are predictors for cognitive reappraisal. Expressive suppression is predicted from both the mother’s positive response for anger (b = −0.28, *p* < 0.01) and happiness (b = −0.38, *p* < 0.001) and negative responses for anger (b = 0.11, *p* < 0.01) and happiness (b = 0.14, *p* < 0.01). 

*Path Model of Anger*. We studied the mediating effects of cognitive reappraisal and expressive suppression between anger socialization and distress (Figure 1 and Figure 2).

Additionally, we investigated emotion regulation mediating effects between parental emotion socialization and psychological distress. 

The relation between the negative emotion socialization of anger and distress is mediated by emotion regulation strategies (expressive suppression, cognitive reappraisal). *Hypothesis 1* is partially confirmed. From all four paths analyzed, we found one mediation in the relation between the maternal negative socialization of anger and distress, occurring through expressive suppression. From our parallel mediation model (Figure 1), which has no supportive emotion socialization relating to anger IV, one of them does not meet the statistical assumptions [36,37] in the model regarding negative anger socialization from the father; IV does not predict M_1_ (cognitive reappraisal) and M_2_ (expressive suppression). The results from our parallel mediation analysis indicate that the maternal negative socialization of anger is indirectly related to psychological distress through its relationship with expressive suppression. In a parallel mediation model, a 95% bias-corrected confidence interval based on 5000 bootstrap samples was indicated as an indirect effect of expressive suppression between the mother’s negative response to anger (b = 0.07, SE = 0.03), CI (0.003 to 0.13) and psychological distress. By contrast, the indirect effects of cognitive reappraisal included zero (−0.01 to 0.08). The relation between the positive emotion socialization of anger (rewarding response) and distress is mediated by emotion regulation strategies (expressive suppression, cognitive reappraisal). *Hypothesis 2* is partially confirmed. From all four paths analyzed, we found two mediations in the relation between the maternal positive socialization of anger and distress, occurring through expressive suppression and cognitive reappraisal. The relation between the father’s emotional response to anger and distress is not mediated by emotion regulation strategies. The model (Figure 2) meets the statistical conditions to conduct parallel mediation models. Our results indicate two significant indirect effects. The maternal reward of anger is indirectly related to psychological distress through cognitive reappraisal (b = −0.21, SE = 0.09), CI (−0.39 to −0.02) and expressive suppression (b = −0.21, SE = 0.08), CI (−0.30 to −0.04). We can therefore say, with 95% confidence, that the indirect effect is negative in all these cases. In this model, we do not have any indirect effects of emotion regulation strategies between the father’s response and distress. 

*Path Model of Happiness.* We studied the mediating effects of cognitive reappraisal and expressive suppression between happiness socialization and distress (Figure 3 and Figure 4).

The relation between negative parental emotion socialization of happiness (punishment and neglect) and distress is mediated by emotion regulation strategies (expressive suppression, cognitive reappraisal). *Hypothesis 3* is partially confirmed. From all four paths analyzed, we found two mediations in the relation between the maternal negative socialization of happiness and distress, occurring through expressive suppression and cognitive reappraisal. Fathers’ negative responses to happiness are directly related with adolescent distress. From the two parallel mediation models we used (Figure 3), one of them does not meet the statistical assumptions of developing parallel mediations, like the IV unsupportive emotion socialization of happiness; in the model with the paternal negative socialization of happiness, IV did not predict M_1_ (cognitive reappraisal) and M_2_ (expressive suppression). We found two significant indirect effects of cognitive reappraisal (b = 0.09, SE = 0.04), CI (0.006 to 0.19) and expressive suppression (b = 0.07, SE = 0.03), CI (8.53 × 10^4^ to 0.14) between the mother’s response and distress.

The relation between the positive emotion socialization of happiness and distress is mediated by emotion regulation strategies (expressive suppression, cognitive reappraisal). *Hypothesis 4* is partially confirmed. From all four paths analyzed, we found two mediations in the relation between the maternal positive socialization of happiness and distress, occurring through expressive suppression and cognitive reappraisal. Fathers’ positive responses to happiness are directly related with adolescent distress. In the model, we have nonsignificant relations between the father’s response in relation to M_1_ (cognitive reappraisal) and M_2_ (expressive suppression). The model has two significant indirect effects of cognitive reappraisal (b = −0.20, SE = 0.09), CI (−0.37 to −0.02) and expressive suppression (b = −0.25, SE = 0.09), CI (−0.043 to −0.06) between the mother’s socialization of happiness and distress.

## 6. Discussion

This study examined the mechanisms underlying the association between parental emotion socialization during childhood and adolescents’—as well as emerging adults’— psychological distress. The results show that not only does emotion socialization have a direct and significant effect on distress, but there are also mechanisms, like emotion regulation strategies, that mediate the relation between parental responses to emotions during the childhood period and adolescents’/emerging adults’ distress. 

Our first significant finding is in accordance with the tripartite model of parental influences [1], which emphasizes family context impact on emotional development through three pathways: the parenting style, the emotional climate and the emotional quality of marital relationships. In our research, the results validate a path model of anger and happiness, which states that a parent’s emotion socialization style and supportive (rewarding) or unsupportive (punishment and neglecting) behavior has a significant influence on a child’s emotional development. When a parent reacts to the anger or happiness of their child in a positive or negative way, he/she creates predictive relations with emotion regulation and distress. Parents’ reactions shape, over time, the ability to deal with emotions from childhood to adolescence and emerging adulthood [13]; these findings are consistent with previous studies [24,25,38]. Unsupportive and cold parental behavior is related to nonadaptive emotion regulation development, even when temperamental dispositions are positive [32]. Emotion regulation abilities and healthy emotional development is significantly better when children receive a high level of positive parenting [39]. 

According to recent research [22], our results confirm the importance of both positive and negative emotion socialization strategies for concrete emotions (e.g., anger and happiness) by the same number of predictive relationships for supportive and unsupportive strategies. 

As such, a second important idea is that positive parenting, although less researched, takes places in the furnishing of individual or interpersonal resources throughout adolescence and emerging adulthood, which are used to regulate emotional experiences and to protect emotional disorders like depression, anxiety and post-traumatic stress disorder [40], in order to grow a sense of personal value and even marital satisfaction [41,42]. Recent neuroimage studies have reflected the connection between positive parent emotion socialization and emotion regulatory brain networks, especially in the primary regions of the prefrontal cortex [43]. 

A third important idea to emphasize is that maternal influence is more present and more evident in the outcome of children in the literature [44,45,46] and in our research. We found predictive relationships for only the mother in relation to emotion regulation strategies and distress. The father’s emotional response is a predictive factor for distress in relation to concrete emotions, especially happiness. Our results are in accordance with the findings from the literature, which indicate that fathers show more happiness and a greater variety of positive and negative emotions than mothers [45]. Perhaps this is due to the fact that fathers have a stronger role when it comes to play and recreational activities, in contrast to mothers, who spend a lot of time engaged in caregiver activities [47]. But, at the same time, the father retains many stereotypical norms of emotion socialization [29], a greater influence in their daughters’ psychopathology [22] and even a long-term influence [13]. Therefore, fathers and mothers have differentiated and specific contributions to the emotional growth of their children [22,28], and this can be an argument for the design of parenting courses that can be projected to take into account gender differences. From an emotion regulation theoretical perspective, one of the key findings of this study is the statistical evidence and explanatory mechanisms of emotion regulation strategies, like protective factors between parental influences and distress. 

The fourth idea of this investigation is that cognitive reappraisal and expressive suppression are emotion regulation strategies, present as mediators between both supportive and unsupportive emotion socialization strategies. A rewarding response of the mother, when it comes to a child’s expression toward anger and happiness, is mediated by expressive suppression and cognitive reappraisal in four mediation paths. A negative response of the mother to their child’s anger and happiness is indirect, related to distress by expressive suppression and cognitive reappraisal in three model paths. 

These results are concordant with a recent study that identifies indirect pathways from parental over-control to major depressive disorders, general anxiety disorders, suicidal thoughts and self-harm via expressive suppression [33]. Concerning cognitive reappraisal, similar research [13] reported, as an unanticipated finding, that neither conflict between offspring and the mother nor conflict between offspring and the father was related to their cognitive reappraisal as emerging adults. Despite this, it is important to note that the impact of parental influences loses its power in emotional disorder issues (e.g., distress, general anxiety disorders or suicidal thoughts) when emotion regulation strategies are mediators [33]. Understanding early emotional experiences is critical for maintaining a lifespan perspective on healthy development, but early experiences per se might not be essential when it comes to answering proximal questions about adolescent, emerging adult and adult emotional functioning [48]. 

Our results add to evidence in the recent literature on emotion socialization [22], which state that parental emotion socialization is indirectly associated with adolescent and emerging adult distress through emotional processes, specifically through emotion regulation strategies. These results are all the more significant, as there are not many analyses in the literature concerning the father–child relationship [44], and because there is evidence demonstrating that different personal choices can transform initial emotional experiences with caregivers and give them a new and assumed direction. 

This study supported the Tripartite Model of the Impact of the Family on Children’s Emotion Regulation and Adjustment [1]. From this theoretical framework, validated in our research, the results are significant in relation to preventative intervention programs for the psychological growth of adolescents. From the perspective of significant relations between parental emotion socialization in a child, adolescent and emerging adult’s emotion regulation and distress, it is evident that emotion socialization parenting programs represent an effective method for the prevention of adolescent difficulties, especially related to emotional responses in a social context. If parents improve their emotion socialization abilities as relates to specific emotions, this will be related to child emotion regulation and psychological wellbeing in turn [49]. Also, it is important to evidence that the parental socialization of emotions has significant indirect influence on an adolescent’s distress, especially as concerns the mother’s response. Of more practical importance is the relation between emotion socialization and distress in the proximal influence of an adolescent’s ER. Given the relation between parental emotion socialization and distress, which is mediated by an adolescent’s emotion regulation, an increase in ER abilities could be a protective factor in relation to a dysfunctional family context [50]. 

## 7. Limitation and Future Directions 

There are some limitations that give us a realistic view and open new directions. First, we have collected data from the self-report only, and thus can carry the risk of mono-method bias. For the future, this bias should be considered by including assessments from parents. Second, the data were collected retrospectively, by measuring past emotional memories that could have been affected by forgetting or even by more positive current attitudes. Therefore, conducting a longitudinal investigation constitutes a difficulty for future work. At the same time, there are authors who consider [51] self-reporting measures of emotion socialization as being the single way of collecting data about parental responses to negative emotions in most cases. Concerning this tendency to express a positive attitude, adolescents tend to appreciate parental emotion socialization more favorably, even idealistically, from a desire to avoid negative emotions associated with perceived relationships with parents [52]. Third, concerning the group of subjects, we studied a large group, who were balanced from the perspective of age but unbalanced by gender grouping (girls predominantly numerically) and not representative of most adolescents and emerging adults. For future research, we intend to expand the sample size and the age group of both pre-adolescents and adults, in order to have continuous perspectives of emotion socialization, especially because maturity brings distance and more objectivity, implicitly in recognition of negative aspects [52]. 

## 8. Conclusions

The mother’s and father’s positive responses to anger and happiness are significant negative predictors of distress; the negative responses of both parents to happiness and the mother’s negative response to anger—but not the father’s—are significant positive predictors of distress. The findings also provide support for the mediating role of expressive suppression and cognitive reappraisal for the mother’s positive response to both anger and happiness, and also for the mother’s negative response to the child’s expression of happiness. None of the father’s responses—positive or negative, in relation to anger or happiness—are mediated by emotion regulation strategies in relation to distress. The findings have some important theoretical and clinical implications for distressed adolescents and emerging adults. 

## Figures and Tables

**Figure 1 healthcare-11-02620-f001:**
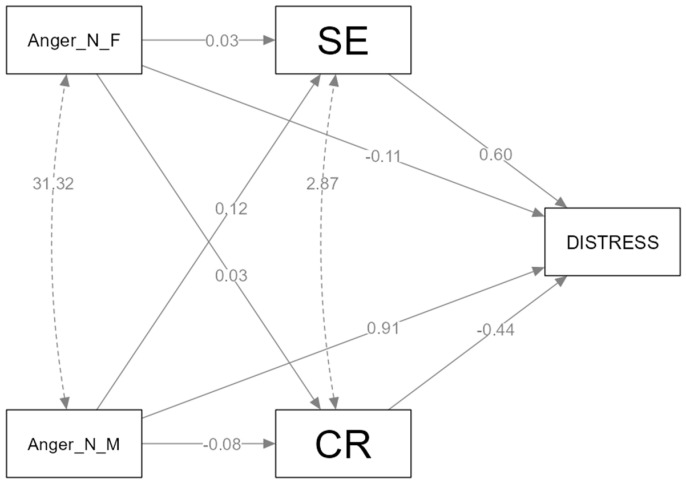
The mediating effect of expressive suppression and cognitive reappraisal between negative anger socialization and distress.

**Figure 2 healthcare-11-02620-f002:**
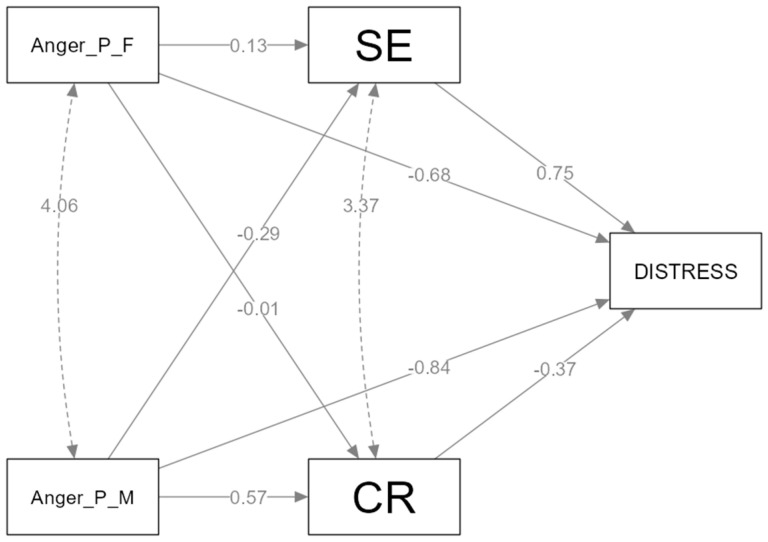
The mediating effect of expressive suppression and cognitive reappraisal in positive anger socialization and distress.

**Figure 3 healthcare-11-02620-f003:**
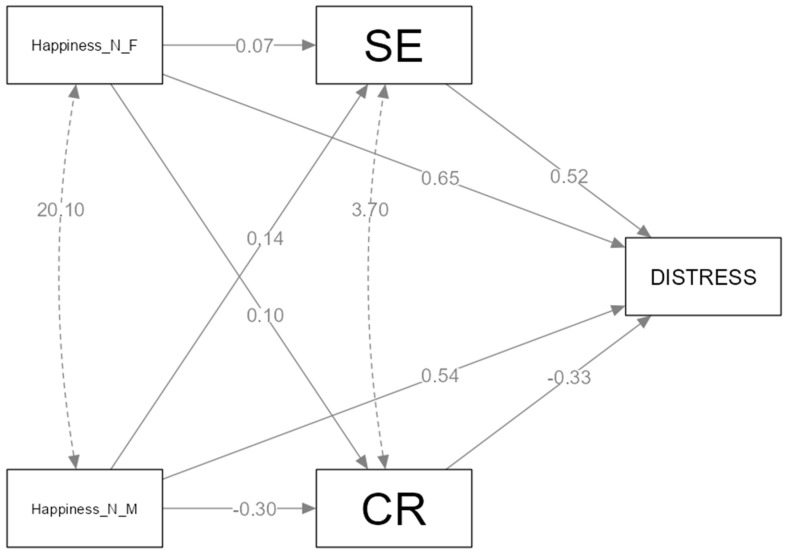
The mediating effect of expressive suppression and cognitive reappraisal between negative happiness socialization and distress.

**Figure 4 healthcare-11-02620-f004:**
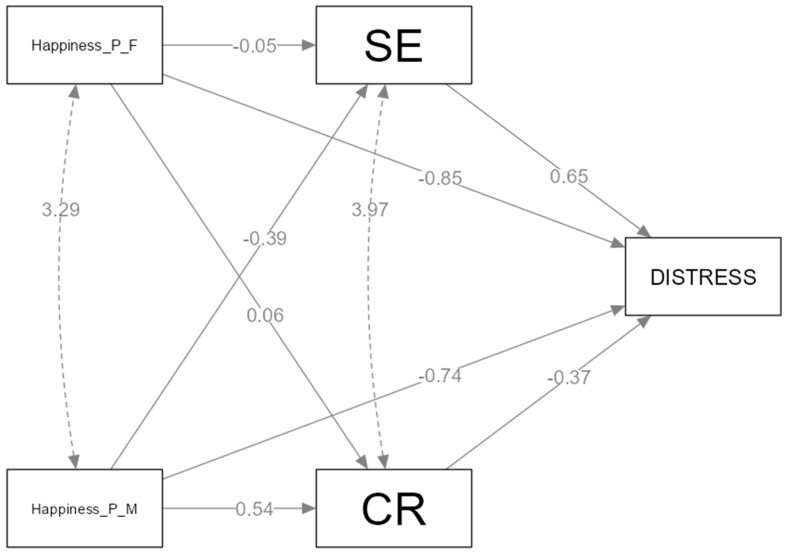
The mediating effect of expressive suppression and cognitive reappraisal between positive happiness socialization and distress.

**Table 1 healthcare-11-02620-t001:** Sample by gender, age, and school level.

		No	Percent
**Gender**	Male	165	33%
	Female	332	67%
**Age**	Adolescent	261	53%
	Emerging adult	236	47%
**School level**	Higher Secondary	222	45%
	University	275	55%
**Total Sample Size**		497	

**Table 2 healthcare-11-02620-t002:** Descriptive statistics.

	*M*	*SD*	Min	Max	Skew	Kurt
Anger_P_M (Anger_positive_mother)	10.66	2.59	3	15	−0.65	0.49
Anger_P_F (Anger_positive_father)	9.23	3.10	3	15	−0.26	−0.62
Anger_N_M (Anger_negative_mother)	6.05	1.87	3	13	0.55	0.22
Anger_N_F (Anger_negative_father)	6.16	1.97	3	14.5	0.62	0.17
Happiness_P_M (Happiness_positive_mother)	12.43	2.37	3	15	−1.26	1.95
Happiness_P_F (Happiness_positive_ father)	11.06	2.85	3	15	−0.71	0.26
Happiness_N_M (Happiness_negative_mother)	4.59	1.83	3	14.5	1.52	2.56
Happiness_N_F (Happiness_negative_father)	5.02	1.80	3	12	0.91	0.38
Cognitive reappraisal (CR)	29.13	5.81	7	42	−0.47	0.38
Expressive suppression (ES)	14.49	4.78	4	27	0.24	−0.35
Distress	56.08	19.65	26	128	0.93	0.54

*M* = mean; *SD* = standard deviation; Min = minimum; Max = maximum; Skew = skewness; Positive is the Reward (mother/father, anger/happiness) measured from 1 (*not at all*) to 5 (*very much*). Negative is the mean between Punishment and Neglect, ranging from 1 (*not at all*) to 5 (*very much*) (mother/father, anger/happiness). Cognitive reappraisal and expressive suppression ranges are from 1 (*strongly disagree*) to 7 (*strongly agree*). Psychological distress ranges from 1 (*not at all*) to 5 (*extremely*).

**Table 3 healthcare-11-02620-t003:** Correlations among the study variables regarding parent socialization emotions and emotional consequences.

	1	2	3	4	5	6	7	8	9	10
1. Anger P_M										
2. Anger P_F	0.53 **									
3. Happiness P_M	0.54 **	0.25 **								
4. Happiness P_F	0.35 **	0.63 **	0.47 **							
5. Anger N_M	−0.35 **	−0.20 **	−0.33 **	−0.20 **						
6. Anger N_F	−0.23 **	−0.33 **	−0.21 **	−0.32 **	0.71 **					
7. Happiness N_M	−0.35 **	−0.13 **	−0.55 **	−0.25 **	0.59 **	0.45 **				
8. Happiness N_F	−0.26 **	−0.35 **	−0.39 **	−0.52 **	0.48 **	0.62 **	0.67 **			
9. CR	0.24 **	0.09 *	0.22 **	0.10 *	−0.11 *	−0.07	−0.19 **	−0.11 *		
10. ES	−0.11 **	−0.01	−0.22 **	−0.14 **	0.18 **	0.12 **	0.20 **	0.19 **	0.08	
11. Distress	−0.21 **	−0.17 **	−0.20 **	−0.20 **	0.31 **	0.25 **	0.30 **	0.34 **	−0.11 *	0.19 **

** *p*< 0.01, * *p* < 0.05; Anger P_M = Anger Positive response from the Mother; Anger P_F = Anger Positive response from the Father; Happiness P_M = Happiness Positive response from the Mother; Happiness P_F = Happiness Positive response from the Father; Anger N_M = Anger Negative response from the Mother; Anger P_F = Anger Positive response from the Father; Happiness P_M = Happiness Positive response from the Mother; Happiness P_F = Happiness Positive response from the Father; CR = Cognitive Reappraisal; ES = Expressive Suppression.

**Table 4 healthcare-11-02620-t004:** The findings from the parallel mediation models (unstandardized).

Independent Measures (IV)	Dependent Measure (DV)	Cognitive Reappraisal (M_1_)											
		Total Effect		Direct Effect		IV → M_1_		M_1_ → DV		Indirect Effect		95% CI	
		*C*		*c*’		*a1*		*b1*		*a1* × *b1*			
		b	SE	b	SE	B	SE	b	SE	b	SE	BootLLCI	BootULCI
Anger_P_M	Distress	−1.26 **	0.38	−0.83 *	0.38	0.56 ***	0.11	−0.37 **	0.14	−0.21 *	0.09	**−0.39**	**−0.02**
Anger_P_F	Distress	−0.58 *	0.31	−0.68 *	0.31	−0.00	0.09	−0.37 **	0.14	0.00	0.03	−3.43	0.06
Anger_N_M	Distress	1.01 ***	0.18	0.91 ***	0.18	−0.07	0.05	−0.44 **	0.13	0.03	0.02	−0.01	0.08
Anger_N_F	Distress	−0.10	0.17	−0.10	0.17	0.03	0.05	0.44 **	0.13	−0.01	0.02	−0.06	0.03
Happiness_P_M	Distress	−1.19 **	0.40	−0.73 *	0.40	0.53 ***	0.11	−0.37 **	0.14	−0.20 *	0.09	**−0.37**	**−0.02**
Happiness_P_F	Distress	−0.90 **	0.33	−0.85 **	0.32	0.05	0.09	−0.37 **	0.14	−0.02	0.03	−0.09	0.05
Happiness_N_M	Distress	0.71 ***	0.21	0.54 **	0.21	−0.29 ***	0.06	−0.33 **	0.14	0.09 *	0.04	**0.006**	**0.19**
Happiness_N_F	Distress	0.65 **	0.20	0.65 **	0.20	0.10	0.06	−0.33 **	0.14	−0.03	0.02	−0.08	0.01
**Independent Measures (IV)**	**Dependent Measure (DV)**	**Expressive Suppression (M_2_)**											
		**Total Effect**		**Direct Effect**		**IV →** **M_2_**		**M_2_ →** **DV**		**Indirect Effect**		**95% CI**	
		** *C* **		***c*’**		** *a1* **		** *b1* **		***a2* × *b2***			
		**b**	**SE**	**b**	**SE**	**B**	**SE**	**b**	**SE**	**b**	**SE**	**BootLLCI**	**BootULCI**
Anger_P_M	Distress	−1.26 **	0.38	−0.83 *	0.38	−0.28 **	0.09	0.74 ***	0.17	−0.21 **	0.08	**−0.30**	**−0.04**
Anger_P_F	Distress	−0.58 *	0.31	−0.68 *	0.31	0.12	0.07	0.74 ***	0.17	0.09	0.06	−0.02	0.21
Anger_N_M	Distress	1.01 ***	0.18	0.91 ***	0.18	0.11 **	0.04	0.59 ***	0.17	0.07 *	0.03	**0.003**	**0.13**
Anger_N_F	Distress	−0.10	0.17	−0.10	0.17	0.02	0.04	0.59 ***	0.17	0.01	0.02	−0.03	0.06
Happiness_P_M	Distress	−1.19 **	0.40	−0.73 *	0.40	−0.38 ***	0.09	0.65 ***	0.17	−0.25 **	0.09	**−0.43**	**−0.06**
Happiness_P_F	Distress	−0.90 **	0.33	−0.85 **	0.32	−0.04	0.08	0.65 ***	0.17	−0.03	0.05	−0.13	0.07
Happiness_N_M	Distress	0.71 ***	0.21	0.54 **	0.21	0.14 **	0.05	0.51 **	0.17	0.07 *	0.03	**8.53 × 10^−4^**	**0.14**
Happiness_N_F	Distress	0.65 **	0.20	0.65 **	0.20	0.07	0.05	0.51 **	0.17	0.03	0.02	−0.02	0.09

* *p* < 0.05; ** *p* < 0.01; *** *p* < 0.001. Anger P_M = Anger Positive response from the Mother; Anger P_F = Anger Positive response from the Father; Happiness P_M = Happiness Positive response from the Mother; Happiness P_F = Happiness Positive response from the Father; Anger N_M = Anger Negative response from the Mother; Anger P_F = Anger Positive response from the Father; Happiness P_M = Happiness Positive response from the Mother; Happiness P_F = Happiness Positive response from the Father; CR = Cognitive Reappraisal; ES = Expressive Suppression. Bolded values do not include zero, indicating a significant indirect effect.

## Data Availability

The data presented in this study are available on request from the corresponding author. The data are not publicly available due to ethical issues.

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
