# Peer review of "Do Emotion Regulation Strategies Mediate the Relationship of Parental Emotion Socialization with Adolescent and Emerging Adult Psychological Distress?"

_healthcare, 2023, doi:10.3390/healthcare11192620_

Round 1
Reviewer 1 Report
Thank you for the opportunity to review the manuscript.
The work submitted for review examines a topic of great relevance in the field of psychology. The topic is very important, and the conceptual analysis made in the text is quite deep. The literature consulted is quite current but the sample isn´t quite large (which is a limitation for your work.). I would like to thank the efforts by the authors of the manuscript and congratulate them on the work. Overall, the writing is clear, the goals are well described, the introduction should explain the objectives of the study based on the review of the previous literature and the conclusions are properly made and presented. I consider that the constructs proposed in the abstract of the work are quite well explained. Therefore, the manuscript brings significant knowledge of the scientific literature so and still covers existing gaps in the field. On a formal level, the manuscript complies with the requirements of the Journal and references are written in accordance with the regulations of the Journal. The work is ambitious, and the results confirm most of the hypotheses and the relevance and potential of the work is therefore recognized, but this Reviewer considers that several changes are needed to the manuscript is publishable. In this sense, it should better explain the novelty and relevance of the work considering the previous empirical evidence and should better describe the practical implications. The process for selecting participants and the procedure should be better described. The study hypotheses should also be better explained. It should describe the discussion and conclusions of the work better and, above all, update the manuscript references (most should be from the last 5 years). Finally, I wish the Authors the best in continuing this line of research.
Best wishes for Authors.
Author Response
Hello,
We have attached the answer to your recommendations.
Thank you for the useful reviews!
With consideration,

Reviewer 2 Report
1. I think it is a very interesting research topic. In addition, clarifying the limitations of the previous study allowed us to better understand how this study differs from the previous study.
2. However, < 1. Theoretical and conceptual framing> should briefly explain the theory that is the basis of this research, and explain how the framework of this research was formed based on the theory. However, I think the explanation for that part is insufficient. They also usually describe a conceptual framework if a theoretical framework or theory is not sufficient. Therefore, it is necessary to reconsider and revise this part. Finally, I think it would be more appropriate to explain the theoretical framework or conceptual framework in the area of methods.
3. The first part of the research method is somewhat unorganized, so it is not naturally connected and read. It would be good if the contents prior to the research method were described more clearly and concisely by including them as Introduction. Also, at the end of the introduction, it would be good if the purpose of the study was explicitly clarified, and a hypothesis was presented below.
4. In the explanation of research results, use sub-numbers to classify hypotheses, and explain whether they are supported or rejected.
5. In the Discussion, it is hoped that not only the comparative analysis of the results of this study and the results of previous studies, but also the interpretation and expansion of the meaning of this study will be further strengthened.
6. And references older than 10 years were used. All of these parts should be replaced with the latest version of the references.
Author Response

(The authors gave the same response as above.)

Reviewer 3 Report
The objective of the manuscript was to explore the mediator’s role of emotion regulation between emotional experiences and emotional life during adolescence and emerging adulthood
---Specific comments---
1. The abstract is insufficiently described. Authors should include in this section that study is focused on parents and offspring (adolescent and emerging adults). At the end, please delete the last sentence because it does not provide information and replace it with a sentence that summarizes the practical implications of the study.
2. The Introduction section needs revision. Please, explain in more detail the tripartite model of family and the Malatesta-Magai model. Do the studies cited focus on adolescents, emerging adults, or both populations? I recommend authors to review the entire introduction and be more clear on this point. Furthermore, throughout the manuscript, the authors refer to emerging adults. Since the authors opt for this label, I recommend cite to Prof. Arnett when referring emerging adults as well as define emerging adulthood as a developmental stage. Prof. Arnett refers to this period specifically as emerging adulthood and not early adulthood. In fact, several of his publications focus on differentiating and defining emerging adulthood as a unique stage different from others. In one of his early publications “Emerging adulthood: A theory of development from the late teens through the twenties” (reference below). Please, review this aspect in the entire manuscript (for example, page 3). Finally, I recommend authors add “the present study” section instead of “Research questions” following 7th Edition APA style. Please, make sure all objectives and hypothesis are stated in this section and explain the results expected.
3. Reference: Arnett, J. J. (2000). Emerging adulthood: A theory of development from the late teens through the twenties. American psychologist, 55(5), 469.
4. Describe the characteristics of the sample in more detail in terms of all socio-demographic variables analyzed. Maybe include a new table would be useful. I wonder: How many participants are parents and how many are adolescent and emerging adults? What age range are the parents and offspring? What number of participants are sons and what number are daughters? Besides, please, explain the procedure deeply.
5. For all measures used, please indicate whether response scores were summed or averaged to create their composite scores for data analysis and add an example item for every scale or subscale used (specifically for psychological distress scale).
6. In the results section, the presentation of the objectives and hypotheses is not the same as in the "Research Questions” section. So, I recommend that the authors review both sections and follow the same order in both
7. Please follow my recommendations in the Introduction section and modify the discussion section accordingly.
8. At the end of the manuscript, please, add practical implications.
Author Response

(The authors gave the same response as above.)

Round 2
Reviewer 2 Report
Thank you for your hard work in revising the paper to the best of your ability, reflecting the reviewer's comments.
Reviewer 3 Report
The authors were very responsive with regard all my observations. I believe that now the manuscript is ready to be accepted.